# Composition of Strawberry Flower Volatiles and Their Effects on Behavior of Strawberry Pollinators, *Bombus terrestris* and *Apis mellifera*

**Jinjia Liu** [1], **Min Chen** [1], **Weihua Ma** [2], **Lifang Zheng** [1], **Bing Zhang** [1], **Huiting Zhao** [3] and **Yusuo Jiang** [1,*]

1 College of Animal Science, Shanxi Agricultural University, Jinzhong 030801, China
2 College of Horticulture, Shanxi Agricultural University, Taiyuan 030031, China
3 College of Life Science, Shanxi Agricultural University, Jinzhong 030801, China
* Correspondence: jiangys-001@163.com; Tel.: +86-0354-6285-990

**Abstract:** Strawberries are popular fruits around the world, and their yield and fruit quality rely on pollination by honey bees and bumblebee colonies. Both bee species have their own advantages in strawberry pollination. This study investigates the characteristic of strawberry (*Fragaria × ananassa* 'Red Face') flower volatiles and their effects on bee pollinators by (1) detecting the volatile compounds of strawberry flowers by polydimethylsiloxane (PDMS) combined gas chromatography-mass spectrometry (GC-MS), (2) determining whether *Bombus terrestris* or *Apis mellifera* showed antennae responses to certain compounds of strawberry flower volatiles by an electroantennography test (EAG), and (3) testing whether these compounds could elicit a corresponding behavioral response in bees. The results showed that (1) there were 38 chemical compounds in 'Red Face' volatiles with 7 types, most of which were known to be generally emitted by flowers but also have some compounds that have not been reported in strawberry flowers; (2) *B. terrestris* and *A. mellifera* had strong EAG responses to several compounds, respectively, especially to ethyl benzoate, (Z)-3-hexenyl propionate, (Z)-3-hexenyl acetate, benzeneacetaldehyde and melonal; and (3) both bee species showed significant avoidance behaviors to four tested compounds, especially the *B. terrestris*. Flower volatiles of strawberry 'Red face' were different from other strawberry varieties that have been reported; some of these electrophysiologically active compounds could cause antennal potential responses in bees, as well as behavioral responses. Our study supports the idea that strawberry flower volatiles are one of the factors influencing bee foraging decisions and provides a reference for formulating more reasonable bee pollination to improve strawberry fruit quality.

**Keywords:** strawberry flower volatiles; *Bombus terrestris*; *Apis mellifera*; EAG response; behavior response

## 1. Introduction

Strawberry (*Fragaria × ananassa* Duch.) belongs to the family Rosaceae and genus Fragaria, and its fruit has excellent edible value. In nature, strawberries rely on wind and insect vectors for pollination, but to achieve high yields to meet market demand, large quantities of strawberry varieties are cultivated in greenhouses [1]. The closed or semi-closed environment hinders the entry of wild insect pollinators and affects strawberry pollination, and the current production is mostly artificial or bee pollination [2].

*F. × ananassa* 'Red Face' is a widely cultivated strawberry variety in China. Honey bees are the dominant managed pollinator worldwide, and commercial bumblebees are also of increasing importance. As superior pollinators in facility crops, these two bee species are commonly used in strawberry pollination in greenhouses [3–5]. Not only do bee pollinators enhance fruit yield, they can also improve fruit quality and reduce the deformed fruit rates of strawberries [1,6,7]. *Apis mellifera* and *Bombus terrestris* are able to increase strawberry production by 50.12% and 41.33% [8]. The rate of deformed fruit of strawberries pollinated by *A. mellifera* and *B. lucorum* was only 17.17% and 11.52%, while the content of fruit vitamin

C reached 0.60 mg/g and 0.67 mg/g [9]. The deformed fruit rate of strawberries pollinated by *B. terrestris* was 81.82% lower than that of naturally pollinated strawberries, and the average fruit quality increased by 37.02% [10]. Nevertheless, bumblebees are thought to be more effective in improving the yield and quality of strawberries in facilities than traditional honey bee species [11,12].

In addition to the buzzing pollination of bumblebees being different from honey bees, flower scents of 'Red Face' may also have a great influence on attracting pollinators [13] because flower volatiles have been reported as one of the main drivers for foraging decisions by honey bees, bumblebees and wild bee species [14–16]. Some studies to date have reported on differences between crop varieties, including the influence on pollinator attraction of varieties differing in flower volatiles [17–20]. It is necessary to clarify whether the volatiles of 'Red Face' flowers differ from those of other varieties because the content and composition of flower volatiles can vary due to genetic differences among subspecies and plant populations at different locations [17,21–24].

Differences in flower volatile compounds among strawberry varieties mediate their attractiveness to pollinators, and peats have been reported [18,23]. *B. terrestris* exhibits a preference for 'Sonata' due to these volatiles being 'less repellent' instead of 'more attractive' than 'Elsanta' [18]. Females of *Osmia bicornis* are also much more abundant flower visitors on 'Sonata' compared to 'Honeoye' and 'Darselect' because all compounds are emitted in the highest quantities by 'Sonata', except nonanal, benzaldehyde, (Z)-3-hexenyl acetate and geranyl acetone [23]. Strawberry blossom weevil (*Anthonomus rubi*) prefers scents of flowering strawberry, but they cannot distinguish scents from *F. × ananassa* and *F. vesca* [17]. All these data show that strawberry floral scent can affect the foraging decisions of pollinators. Nevertheless, whether the flower volatiles of 'Red Face' have different attractiveness to honey bee and bumblebee pollinators is still unknown. We have not found studies comparing different pollinators, which may help us understand how strawberries attract pollinators [25,26].

Strawberries rely on bee pollinators to improve the quality and taste of their fruit, but the characteristics of the flowers may affect the bees' foraging preferences. In this study, the flower volatile compounds of strawberry variety *F. × ananassa* 'Red Face' were identified by GC-MS, and we further tested the EAG response and behavior choices of bumblebee *B. terrestris* and honey bee *A. mellifera* to standard compounds, We explored whether strawberry flower volatiles could influence the electrophysiological and behavior responses of pollinators and whether there were bee species differences between *A. mellifera* and *B. terrestris*. The results of this study help understand strawberry flower volatiles as one of the driving factors influencing bee foraging decisions and further provide better strategies for strawberry pollination.

## 2. Materials and Methods

### 2.1. Strawberry Variety and Bee Species

The strawberry variety *F. × ananassa* 'Red Face' was cultivated in Dabai village, Jinzhong City, Shanxi Province. *A. mellifera* and *B. terrestris* are the main pollinators for strawberry in this area (Figure 1). Honey bees used for all experiments were managed in the apiary of Shanxi Agricultural University. Bumblebees were bought from the Woofuntech Bio-Control Company (Hebei, China) and reared in the darkroom until the experiments.

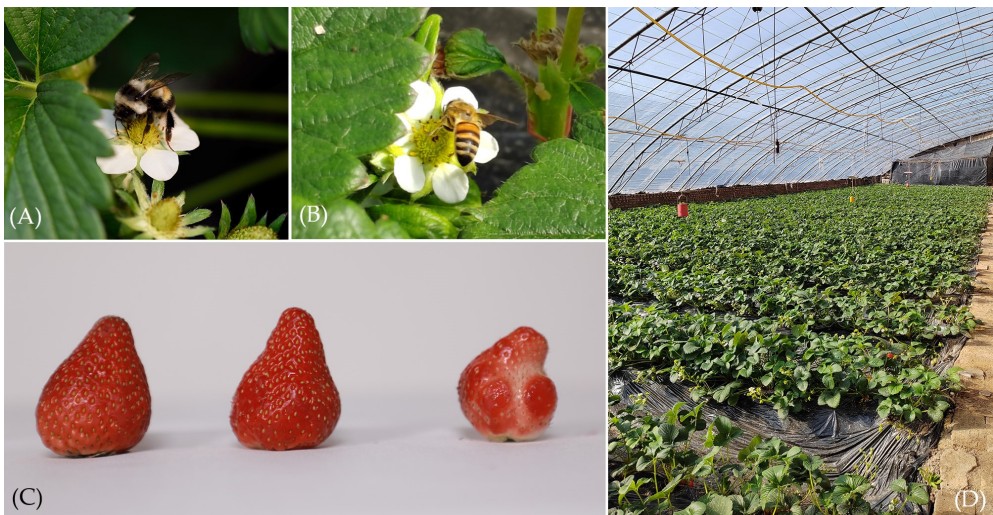

**Figure 1.** Bee pollinating for strawberries in the greenhouse and bee-pollinated fruits: (**A**) bumblebee foraging on a strawberry; (**B**) honey bee foraging on a strawberry; (**C**) left: *A. mellifera* pollinated fruit; middle: *B. terrestris* pollinated fruit; right: naturally pollinated fruit; (**D**) the greenhouse of *F. × ananassa* 'Red Face' (length × width × height: 94 m × 8 m × 4 m).

### 2.2. Flower Volatiles Collection and Detection

Newly opened flowers of 'Red Face' in the greenhouse that have not been pollinated were cut and brought back to the laboratory under 4 °C; 5 g weighted with a balance was put into a 20 mL sample vial and sealed with a vial cap in a 3 mL NaCl solution. Before collection, the sample vial was equilibrated for 0.5 h at a temperature of 50 °C. Then, a 100 μL PDMS fiber (Supelco, St. Louis, MO, USA) was used for sample extraction. After a 0.5 h extraction, the sample fiber was removed from the vial and inserted in the inlet of the tube column for desorption. The temperature of the inlet was 240 °C, and the fiber was desorbed for 5 min.

For qualitative and quantitative analysis of the flower volatile samples, GC-MS (Agilent 6890N-5975B) was used. The GC conditions were as follows: GC column was HP-5MS (0.25 mm × 30 mm × 0.25 μm); the inlet temperature was 240 °C; helium was the carrier gas (purity ≥ 99.99%); the flow rate was 1.0 mL/min. The oven program: the initial temperature was 45 °C, maintained for 5min and then warmed from 45 to 130 °C at a rate of 6 °C/min; it was finally heated from 130 to 240 °C at a rate of 10 °C/min and maintained for 8 min. The injection mode was splitless. For MS, the ionization mode was an electron ion source (EI) with the ion source temperature at 230 °C, and the interface temperature was 250 °C. A full scan was conducted, and the mass scan range was 45–500 *m/z*. For volatile compounds identification, mass spectra of all chromatographic peaks were extracted and matched to the NIST 14 library.

### 2.3. EAG Responses to Volatile Compounds

To test the sensitivity of the EAG responses of bees, standard compounds were diluted in liquid paraffin at six different concentrations (10 μg/μL, 100 μg/μL, 200 μg/μL, 300 μg/μL, 400 μg/μL and 500 μg/μL). All standard compounds were supplied by commercial companies (Table 1). Not all of the compounds identified were obtained at the time of our study. Honey bee foragers were randomly caught from different bee hives and starved for 1 d before the tests. The method for the EAG was based on reference and adapted to our requirements of *A. mellifera* and *B. terrestris* [27]. Firstly, the left antenna of an active bee was cut off from the base by a scalpel blade under the asana microscope. Then, a reference electrode was inserted into the base. Finally, a small opening was cut at the top of the antenna. A recording electrode was inserted into this incision. The reference and the

recording glass electrodes were filled with Ringer's solution (8.0 g/L NaCl, 0.4 g/L KCl, 0.4 g/L CaCl$_2$) and contacted with Ag/AgCl wires.

**Table 1.** Information of standard compounds used in the EAG tests and the Y-tube experiments to *A. mellifera* and *B. terrestris*.

| Compounds | Purity (%) | Origin |
|---|---|---|
| Ethyl benzoate | >99.5 | Aladdin [1] |
| Methyl salicylate | ≥99.5 | Aladdin |
| Ethyl salicylate | 99.0 | Aladdin |
| Ethyl palmitate | ≥99.0 | Aladdin |
| (Z)-3-Hexenyl acetate | 98.0 | Aladdin |
| (Z)-Hex-3-enyl benzoate | 97.0 | Aladdin |
| 2,2,4-Trimethyl-1,3-pentadienol diisobutyrate | 98.5 | Aladdin |
| Ethyl cinnamate | 99.0 | Macklin [2] |
| Cis-3-Hexenyl isovalerate | 97.0 | Aladdin |
| Ethyl myristate | ≥98.0 | Aladdin |
| (Z)-3-Hexenyl propionate | 97.0 | Macklin |
| Ethyl dodecanoate | 99.0 | Aladdin |
| Staflex BOP | ≥95.0 | Macklin |
| Benzyl benzoate | >99.0 | Aladdin |
| Linalool | 98.0 | Aladdin |
| Benzyl alcohol | ≥99.5 | Aladdin |
| Phenylethyl alcohol | ≥99.5 | Aladdin |
| Nonadecane | 98.0 | Macklin |
| 1-Chlorododecane | 98.0 | Aladdin |
| 1-Chlorotetradecane | 98.0 | Aladdin |
| β-Ionone | 97.0 | Aladdin |
| Methyleugenol | 98.0 | Aladdin |
| 1,3-Ditert-butylbenzene | >98.0 | Aladdin |
| Eugenol | 99.0 | Aladdin |
| Melonal | 80.0 | Aladdin |
| Benzeneacetaldehyde | 95.0 | Aladdin |

[1] Aladdin: Shanghai, China; [2] Macklin: Shanghai, China.

The DC potential change of the antenna was transmitted through a combi-probe (Universal AC/DC probe; Syntech, the Netherlands) and recorded with EAG 2000 software (Syntech, Hilversum, the Netherlands). An air stimulus controller (CS55; Syntech, Hilversum, the Netherlands) was used to provide the purified air with testing compound odors, and the airflow was at a rate of 18 L/h. The position of the testing antenna was adjusted by micro-manipulator arms (MP-15; Syntech, Hilversum, the Netherlands) so that it was in the center of the airflow carrying the compound passing through. For every concentration, the plus duration for each stimulation was 0.5 s, and there was a 30 s interval between two stimulations. The time intervals between each compound were 1 min. In order to avoid olfactory adaption, the stimulation order was from low concentration to high concentration. The responses to liquid paraffin were the blank control of the antenna before and after all stimulation. To obtain stable replicate data, we used at least 6 *A. mellifera* individuals and 4 *B. terrestris* individuals for each compound.

### 2.4. Y-Tube Olfactometer Behavior Tests to Volatile Compounds

Based on the EAG tests, four standard compounds were used for Y-tube behavioral tests. These compounds were also diluted in liquid paraffin at six different concentrations (10 μg/μL, 100 μg/μL, 200 μg/μL, 300 μg/μL, 400 μg/μL and 500 μg/μL). To avoid the effects of phototaxis, an odor-free room with a red light was prepared, and the temperature was maintained at around 25 °C. The parameters of our Y-tube olfactometers: stem 25 cm, arms 18 cm, at an angle of 75°, the internal diameter was 3.0 cm. The arms of the Y-tube were connected to the odor supply cylinder, the gas washing cylinder and the activated charcoal filter, respectively, and finally connected to an air pump. A 3 × 1.5 cm$^2$ filter paper

strip that was applied a 10 μL dose of each compound/liquid paraffin was put into the odor supply cylinder for a 30 s evaporation. Then, the cleaned airflow (500 mL/min) with odor was pumped from the two Y-tube arms to the stem for bee individuals to make a choice.

Honey bee foragers were randomly collected from the entrances of the beehive. Bumblebee workers were selected from the colony a day in advance and placed in feeding boxes. All bee individuals were allowed to choose no more than 5 min in the Y-tube. Bees were recorded as making choices when they moved 10 cm of the length of one arm and stayed there for at least 5 s. The positions of control and treatment compounds were exchanged after every 5 bees were tested, and for every 10 individuals tested, a clean Y-tube was replaced. Each concentration of the compound was tested in two bee species with 30 bee individuals, respectively. The differences in the choices of bees were analyzed.

### 2.5. Statistical Analyses

The ratio of individual component peak area to total peak areas was used to represent the content of each volatile compound. Due to individual differences between bees, the EAG response of the compound relative to the control (liquid paraffin) was used as the actual EAG response of bees to different concentrations of different compounds for statistical analysis. For EAG response differences between bee species and differences among applied concentrations of the same compound, the data were firstly tested for normal distribution and then analyzed with non-parametric tests (Kruskal–Wallis test). For behavioral preference to a different compound, the Chi-squared test ($\chi^2$) was used to analyze the difference in choice. Bee individuals that did not tend to any compounds were excluded from the analysis. IBM SPSS Statistics for Windows, version 26 (IBM Corp., Armonk, NY, USA) was used for all data analysis, and OriginPro for Windows, version 2020 (OriginLab Corp., Northampton, MA, USA) were used for drawing.

## 3. Results

### 3.1. Types and Contents of Volatile Compounds of Strawberry Flowers

The volatile compounds of strawberry flowers mainly include esters, alcohols, alkanes, ketones, ethers, aromatics and aldehydes (Figure 2). Among these 38 volatile compounds, the number of esters were the most with 15 compounds, followed by alcohols with 6 compounds, then alkanes with 5 compounds, aromatics with 5 compounds, aldehydes with 4 compounds, ketones with 2 compounds, and only 1 ether. Esters accounted for 18.11% of total volatiles, with ethyl benzoate (6.27%) accounting for the highest content in esters, followed by methyl salicylate (3.08%); alcohols accounted for 10.92%, with linalool (9.12%) accounting for the highest content; alkanes accounted for 3.48%, with nonadecane (1.72%) accounting for the highest content; ketones accounted for 3.7%, with *N*-succinimidyl benzoate (2.79%) and β-violet (0.91%); aromatics accounted for 15.74%, with eugenol (9.65%) accounting for the highest content, followed by 1,3-ditert-butylbenzene (4.11%); aldehydes accounted for 24.91%, with 2,6-dimethyl-5-heptaneal (21.25%) accounting for the highest content in aldehydes and ethers, of which methyl ether accounted for 23.14%, which was also the highest of total volatiles (Table 2).

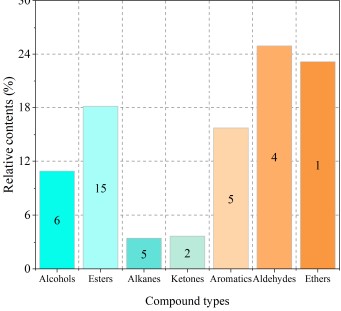

**Figure 2.** Contents of compound types in strawberry volatiles is presented with columns in different colors. The figure on the column means the number of compounds of this type.

**Table 2.** Number and relative content (% of each compound) of the compounds identified in 'Red Face' flower volatiles.

| Types/No. | Compounds | CAS Number | RT | Relative Content (%) |
|---|---|---|---|---|
| Alcohols | | | | |
| 1 | Linalool | 78-70-6 | 13.601 | 9.12 |
| 2 | Benzyl alcohol | 110-51-6 | 12.246 | 1.23 |
| 3 | Phenylethyl alcohol | 60-12-8 | 17.134 | 0.05 |
| 4 | Erythro-1-phenyl-1,2-propanediol | 1075-04-3 | 13.01 | 0.02 |
| 5 | Bicyclo[3.3.1]non-2-en-9-ol, syn- | 19877-78-2 | 17.967 | 0.34 |
| 6 | Ledol | 577-27-5 | 31.641 | 0.16 |
| Esters | | | | |
| 7 | Ethyl benzoate | 93-89-0 | 16.412 | 6.27 |
| 8 | Methyl salicylate | 119-36-8 | 17.429 | 3.08 |
| 9 | Ethyl salicylate | 118-61-6 | 21.599 | 2.62 |
| 10 | Ethyl palmitate | 628-97-7 | 36.995 | 1.31 |
| 11 | (Z)-3-Hexenyl acetate | 3681-71-8 | 9.71 | 1.23 |
| 12 | (Z)-3-Hexenyl benzoate | 25152-85-6 | 30.292 | 0.77 |
| 13 | 2,2,4-Trimethyl-1,3-pentadienol diisobutyrate | 6846-50-0 | 30.722 | 0.67 |
| 14 | Tris ethyl cinnamate | 4192-77-2 | 28.229 | 0.43 |
| 15 | Cis-3-Hexenyl isovalerate | 35154-45-1 | 19.565 | 0.36 |
| 16 | Ethyl myristate | 124-06-1 | 33.694 | 0.36 |
| 17 | (Z)-3-Hexenyl propionate | 33467-74-2 | 19.368 | 0.31 |
| 18 | Ethyl dodecanoate | 106-33-2 | 30.653 | 0.25 |
| 19 | Staflex BOP | 84-78-6 | 34.859 | 0.13 |
| 20 | Benzyl benzoate | 120-51-4 | 33.452 | 0.09 |
| 21 | Benzyl valerate | 10361-39-4 | 26.434 | 0.23 |
| Alkanes | | | | |
| 22 | Nonadecane | 629-92-5 | 39.455 | 1.72 |
| 23 | 1-Chlorododecane | 112-52-7 | 21.133 | 0.55 |
| 24 | 1-Chlorotetradecane | 2425-54-9 | 28.77 | 0.35 |
| 25 | Heptacosane | 593-49-7 | 42.332 | 0.84 |
| 26 | 1-Chloro-5-methyl hexane | 33240-56-1 | 24.997 | 0.02 |
| Ketones | | | | |
| 27 | N-Succinimidyl benzoate | 23405-15-4 | 8.191 | 2.79 |
| 28 | β-Ionone | 79-77-6 | 28.632 | 0.91 |
| Aromatics | | | | |
| 29 | Methyleugenol | 93-15-2 | 26.818 | 0.33 |
| 30 | 1,3-Ditert-butylbenzene | 1014-60-4 | 20.464 | 4.11 |
| 31 | Eugenol | 97-53-0 | 25.42 | 9.65 |
| 32 | Butylated hydroxytoluene (BHT) | 128-37-0 | 29.183 | 0.23 |
| 33 | 2,4-Ditert-butylphenol | 96-76-4 | 29.338 | 1.42 |
| Aldehydes | | | | |
| 34 | Melonal | 106-72-9 | 4.98 | 21.25 |
| 35 | Benzeneacetaldehyde | 122-78-1 | 11.199 | 2.96 |
| 36 | 1,3,4-Trimethyl-3-cyclohexene-1-carboxaldehyde | 40702-26-9 | 18.64 | 0.57 |
| 35 | (Z)-7-Hexadecenal | 56797-40-1 | 32.3 | 0.13 |
| Ethers | | | | |
| 38 | Dimethyl ether | 115-10-6 | 1.798 | 23.14 |

### 3.2. EAG Response to Volatile Compounds

EAG responses of honey bees and bumblebees to different compounds were different (Figure 3). *B. terrestris* elicited strong antennal responses to methyl salicylate, ethyl benzoate, (Z)-3-hexenyl propionate, (Z)-3-hexenyl acetate, linalool, benzyl alcohol, phenylethyl alcohol, eugenol, methyleugenol and melonal (2,6-dimethyl-5-heptenal) when relative to liquid paraffin. EAG responses of *A. mellifera* were higher than liquid paraffin to staflex BOP, methyl salicylate, ethyl salicylate, (Z)-3-hexenyl propionate, (Z)-3-hexenyl acetate, linalool, benzyl alcohol, phenylethyl alcohol, eugenol, methyleugenol, benzeneacetaldehyde and melonal. Among these compounds, responses of *A. mellifera* to benzeneacetaldehyde were

significantly higher than that of *B. terrestris*, and responses of *B. terrestris* to ethyl benzoate were significantly higher than that of *A. mellifera*.

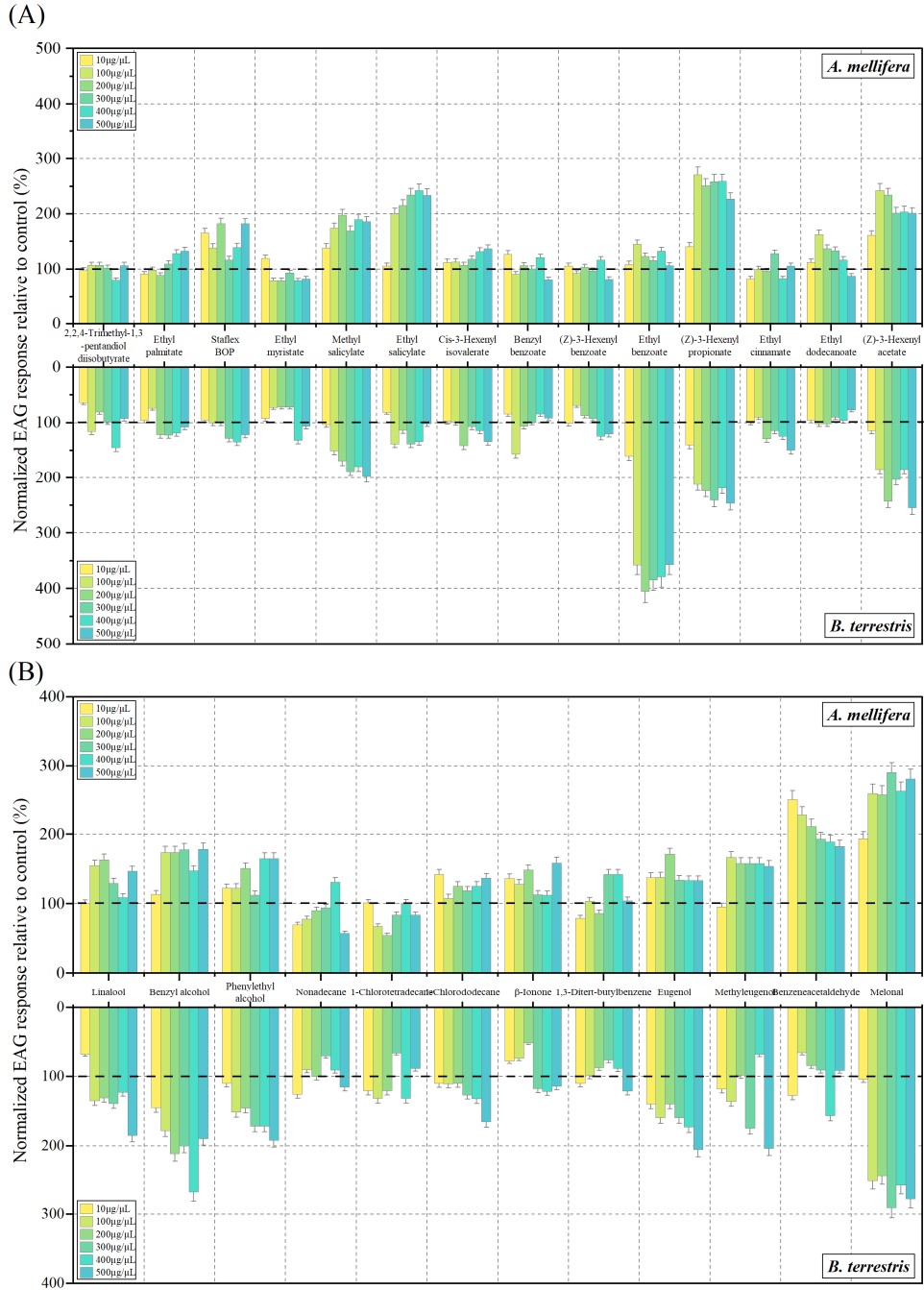

**Figure 3.** EAG responses of bees to volatile compounds: (**A**) EAG responses of bees to esters relative to liquid paraffin; (**B**) EAG response of bees to other compounds relative to liquid paraffin. The 100% black dotted line represents the EAG response of the control liquid paraffin.

EAG responses of bees to the same compounds were also different at concentrations (Table 3). The EAG responses of *A. mellifera* to 18 compounds were significantly different among concentrations, and for most compounds, the EAG response values were not positively or negatively correlated with concentration. For *B. terrestris*, there were 11 compounds with concentration differences in EAG responses, and most of them were positively correlated with concentration.

**Table 3.** Significant differences in EAG responses (mean ± sem.) to different concentrations of the same compound of *A. mellifera* and *B. terrestris*.

| | Mean EAG Response to Different Compounds (%, Diluted in Liquid Paraffin) | | | | | |
|---|---|---|---|---|---|---|
| **Bee Species/Compounds** | **10 µg/µL** | **100 µg/µL** | **200 µg/µL** | **300 µg/µL** | **400 µg/µL** | **500 µg/µL** |
| *A. mellifera* | | | | | | |
| Ethyl benzoate | 8.94 ± 0.19 [b] | 45.41 ± 0.27 [a] | 23.05 ± 0.18 [b] | 15.92 ± 0.16 [b] | 33.04 ± 0.15 [ab] | 6.24 ± 0.11 [b] |
| Ethyl salicylate | 25.89 ± 0.81 [b] | 38.66 ± 0.27 [b] | 57.39 ± 0.17 [ab] | 53.13 ± 0.10 [ab] | 179.38 ± 0.75 [a] | 218.25 ± 0.81 [a] |
| Ethyl palmitate | 8.12 ± 0.01 [c] | 6.29 ± 0.02 [c] | 14.64 ± 0.05 [bc] | 14.04 ± 0.03 [bc] | 30.54 ± 0.10 [ab] | 37.02 ± 0.03 [a] |
| (Z)-3-Hexenyl acetate | 41.84 ± 0.12 [b] | 213.12 ± 0.10 [a] | 219.15 ± 0.27 [a] | 142.29 ± 0.48 [ab] | 145.00 ± 0.50 [ab] | 70.86 ± 0.17 [ab] |
| Tris ethyl cinnamate | 15.24 ± 0.04 [b] | 25.38 ± 0.08 [b] | 23.86 ± 0.02 [b] | 9.08 ± 0.17 [b] | 47.16 ± 0.05 [a] | 13.89 ± 0.04 [b] |
| Cis-3-Hexenyl isovalerate | 4.32 ± 0.01 [b] | 37.30 ± 0.09 [a] | 36.25 ± 0.09 [a] | 17.18 ± 0.08 [ab] | 19.86 ± 0.05 [ab] | 19.19 ± 0.13 [ab] |
| Ethyl myristate | 12.49 ± 0.03 [c] | 12.67 ± 0.01 [c] | 20.03 ± 0.08 [bc] | 28.08 ± 0.11 [bc] | 63.34 ± 0.03 [a] | 43.56 ± 0.07 [ab] |
| (Z)-3-Hexenyl propionate | 27.10 ± 0.06 [b] | 75.84 ± 0.29 [ab] | 59.35 ± 0.16 [ab] | 116.10 ± 0.25 [a] | 103.27 ± 0.31 [ab] | 96.34 ± 0.23 [ab] |
| Ethyl dodecanoate | 51.72 ± 0.10 [a] | 24.43 ± 0.07 [b] | 21.53 ± 0.06 [b] | 26.20 ± 0.08 [b] | 14.12 ± 0.02 [b] | 15.25 ± 0.01 [b] |
| Butyl octyl phthalate | 17.23 ± 0.08 [bc] | 14.73 ± 0.04 [c] | 24.27 ± 0.04 [abc] | 41.65 ± 0.07 [ab] | 45.44 ± 0.10 [a] | 32.89 ± 0.15 [abc] |
| Linalool | 37.06 ± 0.05 [ab] | 30.06 ± 0.07 [b] | 70.78 ± 0.17 [a] | 56.78 ± 0.13 [ab] | 41.52 ± 0.08 [ab] | 42.98 ± 0.03 [ab] |
| Benzyl alcohol | 34.64 ± 0.10 [b] | 34.10 ± 0. 04 [b] | 23.96 ± 0.12 [b] | 142.58 ± 0.22 [a] | 25.54 ± 0.03 [b] | 30.36 ± 0.14 [b] |
| Phenylethyl alcohol | 48.28 ± 0.08 [ab] | 30.00 ± 0.03 [b] | 64.85 ± 0.16 [ab] | 37.66 ± 0.07 [ab] | 73.29 ± 0.08 [a] | 59.06 ± 0.16 [ab] |
| 1-Chlorododecane | 41.46 ± 0.02 [c] | 17.42 ± 0.04 [d] | 24.51 ± 0.05 [cd] | 45.37 ± 0.02 [bc] | 62.96 ± 0.10 [b] | 118.77 ± 0.11 [a] |
| 1-Chlorotetradecane | 26.08 ± 0.07 [c] | 62.02 ± 0.05 [a] | 39.81 ± 0.01 [bc] | 27.42 ± 0.05 [c] | 54.90 ± 0.05 [ab] | 32.22 ± 0.06 [c] |
| β-Ionone | 30.20 ± 0.03 [b] | 10.51 ± 0.06 [b] | 51.23 ± 0.01 [b] | 127.68 ± 0.04 [a] | 56.07 ± 0.27 [b] | 42.24 ± 0.18 [b] |
| Methyleugenol | 3.35 ± 0.01 [b] | 45.42 ± 0.04 [a] | 31.76 ± 0.09 [ab] | 35.05 ± 0.17 [ab] | 28.48 ± 0.10 [ab] | 27.12 ± 0.09 [ab] |
| 1,3-Di-tert-butylbenzene | 8.12 ± 0.02 [b] | 25.09 ± 0.02 [ab] | 22.26 ± 0.03 [ab] | 35.41 ± 0.06 [a] | 24.99 ± 0.05 [ab] | 31.24 ± 0.12 [a] |
| *B. terrestris* | | | | | | |
| Ethyl palmitate | 50.03 ± 0.04 [ab] | 71.82 ± 0.09 [a] | 34.50 ± 0.22 [ab] | 29.06 ± 0.11 [ab] | 27.79 ± 0.10 [ab] | 15.07 ± 0.07 [b] |
| (Z)-3-Hexenyl acetate | 27.42 ± 0.13 [b] | 84.36 ± 0.37 [ab] | 142.34 ± 0.62 [ab] | 108.13 ± 0.45 [ab] | 124.85 ± 0.75 [ab] | 208.97 ± 0.22 [a] |
| Cis-3-Hexenyl isovalerate | 5.51 ± 0.05 [c] | 20.76 ± 0.11 [c] | 42.26 ± 0.10 [b] | 31.28 ± 0.08 [bc] | 31.28 ± 0.06 [bc] | 79.79 ± 0.13 [a] |
| Ethyl myristate | 5.88 ± 0.02 [b] | 7.96 ± 0.03 [b] | 24.03 ± 0.06 [ab] | 23.45 ± 0.10 [ab] | 13.48 ± 0.03 [ab] | 69.02 ± 0.30 [a] |
| Benzyl benzoate | 11.66 ± 0.03 [b] | 63.38 ± 0.27 [a] | 35.92 ± 0.12 [ab] | 7.53 ± 0.03 [b] | 22.36 ± 0.13 [ab] | 36.93 ± 0.05 [ab] |
| Linalool | 42.10 ± 0.08 [ab] | 21.80 ± 0.11 [ab] | 26.96 ± 0.21 [ab] | 1.56 ± 0.01 [b] | 43.61 ± 0.07 [ab] | 84.96 ± 0.40 [a] |
| Benzyl alcohol | 19.71 ± 0.05 [b] | 28.22 ± 0.03 [b] | 79.08 ± 0.42 [ab] | 71.68 ± 0.56 [ab] | 227.13 ± 0.91 [a] | 61.31 ± 0.27 [b] |
| Phenylethyl alcohol | 3.75 ± 0. 01 [b] | 69.68 ± 0.24 [ab] | 74.68 ± 0.28 [ab] | 41.26 ± 0.15 [ab] | 71.53 ± 0.31 [ab] | 92.33 ± 0.25 [a] |
| 1-Chlorotetradecane | 44.44 ± 0.05 [a] | 32.53 ± 0.13 [ab] | 49.11 ± 0.10 [a] | 23.66 ± 0.05 [ab] | 43.85 ± 0.14 [a] | 6.95 ± 0.01 [b] |
| 1,3-Di-tert-butylbenzene | 45.44 ± 0.05 [ab] | 28.22 ± 0.08 [b] | 27.26 ± 0.11 [b] | 62.51 ± 0.08 [a] | 50.49 ± 0.09 [ab] | 59.10 ± 0.10 [a] |
| Eugenol | 33.08 ± 0.07 [b] | 47.28 ± 0.07 [b] | 47.28 ± 0.07 [b] | 81.37 ± 0.21 [ab] | 57.88 ± 0.07 [ab] | 105.66 ± 0.24 [a] |

Different superscript letters ([a], [b], [c]) indicate the significant difference among different concentrations of the same compounds ($p < 0.05$, one-way ANOVA followed by Kruskal–Wallis test). Values are means ± standard error ($n = 6$ for *A. mellifera*; $n = 4$ for *B. terrestris*).

### 3.3. Y-Tube Behavior Tests of Bees

Both bee species showed significant behavior responses to different compounds. For (Z)-3-hexenyl acetate (Figure 4A), the choice percentage of *B. terrestris* was significantly higher than control liquid paraffin at a concentration of 10 µg/µL ($\chi^2 = 4.263$, df = 1, $p = 0.039$), 200 µg/µL ($\chi^2 = 13.235$, df = 1, $p < 0.001$) and 400 µg/µL ($\chi^2 = 9.008$, df = 1, $p = 0.003$), and there were no differences at a concentration of 100, 300 and 500 µg/µL ($p > 0.05$). Differently, the percentage of *A. mellifera* in (Z)-3-hexenyl acetate only had a significant difference at a concentration of 200 µg/µL ($\chi^2 = 9.308$, df = 1, $p = 0.002$). For melonal (Figure 4B), the choice percentage of *B. terrestris* was significantly higher than control liquid paraffin at five concentrations except for 10 µg/µL ($p > 0.05$), and

*A. mellifera* only showed behavior preference at a concentration of 500 µg/µL ($\chi^2$ = 6.00, df = 1, $p$ = 0.014). For (*Z*)-3-hexenyl propionate (Figure 4C), the choice percentage of *B. terrestris* was significantly higher than control liquid paraffin at a concentration of 10 µg/µL ($\chi^2$ = 4.840, df = 1, $p$ = 0.028), 100 µg/µL ($\chi^2$ = 6.760, df = 1, $p$ = 0.009), 200 µg/µL ($\chi^2$ = 4.167, df = 1, $p$ = 0.041) and 500 µg/µL ($\chi^2$ = 13.370, df = 1, $p$ < 0.001). *A. mellifera* only had a significant difference at a concentration of 100 µg/µL ($\chi^2$ = 6.368, df = 1, $p$ = 0.012). For Benzeneacetaldehyde (Figure 4D), *B. terrestris* significantly showed avoidance behavior except at the lowest concentration 10 µg/µL ($\chi^2$ = 0.048, df = 1, $p$ = 0.827), and *A. mellifera* showed no significantly behavioral responses except at a concentration of 400 µg/µL ($\chi^2$ = 9.00, df = 1, $p$ = 0.003).

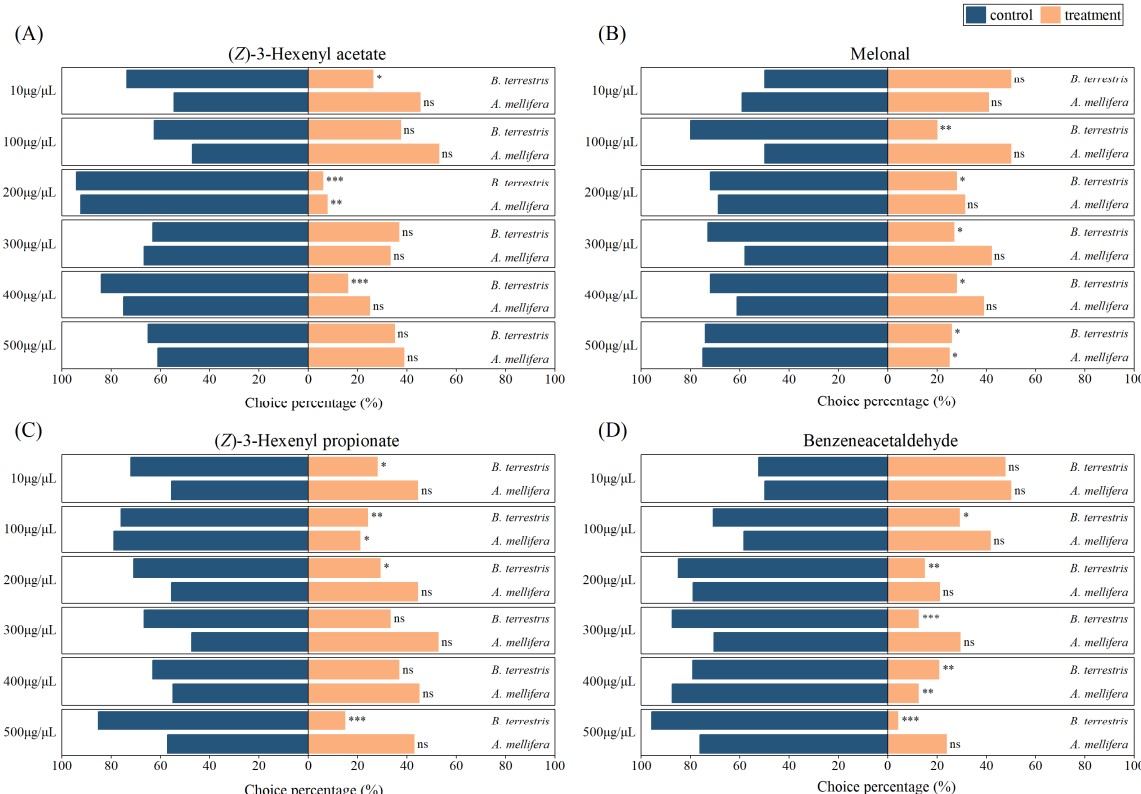

**Figure 4.** Difference in behavior choices of bees to different compounds at six concentrations: (**A**) the behavior choice of bees to (*Z*)-3-hexenyl acetate; (**B**) the behavior choice of bees to melonal; (**C**) the behavior choice of bees to (*Z*)-3-hexenyl propionate; (**D**) the behavior choice of bees to benzeneacetaldehyde. (ns: $p$ > 0.05; * 0.01 < $p$ < 0.05, ** 0.001 < $p$ < 0.01, and *** $p$ < 0.001).

## 4. Discussion

Bumblebees and honey bees are widely used as pollinators in order to enhance fruit yield and quality in strawberries. However, the preferences of bees for certain flower traits can lead to different foraging rates, affecting the intended pollination services, fruit yield and quality [14]. Here, we investigated the importance of strawberry flower volatiles mediating bee–strawberry pollination interactions. The findings of this study revealed that 'Red Face' released different compounds compared with other strawberry varieties, and honey bees and bumblebees had different EAG responses to strawberry flower volatiles, as well as in behavior choices, which may influence the foraging choice of bees to cultivated strawberries. While bumblebees and honey bees do respond differently to strawberry volatiles, it was still hard to explain the overly complex relationship between pollinators and plants because the Y-tube experiment results were inconsistent with bee behavior in the greenhouse. This study provides evidence for strawberry flower volatiles as a factor

that influence the foraging decision of bees for strawberry pollination, which will help to further develop more reasonable strawberry pollination guidelines.

The composition of flower volatiles from several species of strawberries has been reported previously, including *F.* × *ananassa* [18,22,23,28], *F. virginiana* Duchesne [29], *F. vesca* [30] and *F. viridis* Duchesne [31]. We found that dimethyl ether was the major constituent of the floral volatiles released by our samples, contributing 23.14% of the total amount. It contrasts with previous reports that 4-methoxybenzaldehyde (*p*-anisaldehyde) was considered to be the major constituent of the floral volatiles released by *F.* × *ananassa* varieties [17], benzaldehyde was also present in a large amount in flower volatiles of *F.* × *ananassa* varieties 'Darselect', 'Honeoye' [23] and 'Korona' [28]. In another study involving the variety 'Sonata', (*E,E*)-α-farnesene and limonene were reported as the major component of floral volatiles [18,23]. However, in this study, these compounds were not present in variety 'Red Face'.

There was a total of 38 kinds of compounds (seven types: aldehyde, ether, ester, aromatic, alcohol, ketone, and alkane) in 'Red Face' flower volatiles detected in our study, unlike other studies that have reported about the compositions and quantity of the flower volatile compounds emitted by strawberries [17,22,23]. We found more esters (15 compounds), alcohols (6 compounds) and ether (dimethyl ether) in 'Red Face' but also did not find some compounds in our samples. Most of the compounds we found in the current study were not found before, especially dimethyl ether which accounted for the most content and are reported here for strawberries for the first time. Different volatile collection methods may contribute to this discrepancy, SPME provided a higher sensitive technique able to detect more compounds than static and dynamic headspace collections [23]. Variety differences may also exist because it was clear that differences in volatiles among strawberry varieties are common [17,23]. In any case, compounds emitted by strawberry flowers are known to be general, and almost half of the compounds are found among the most frequently emitted flower volatile compounds [32–35].

Antennal responses of *B. terrestris* and *A. mellifera* to several volatile compounds were higher than responses to controls, and the responses differed among most compounds. Honey bees are known to respond to several of the compounds that were found in our study, namely linalool, (*Z*)-3-hexenyl acetate, methyl salicylate, benzyl alcohol and benzeneacetaldehyde [19,36–38]. In addition, melonal and (*Z*)-3-hexenyl benzoate also elicited strong responses that are yet to be reported. Some of these compounds (methyl benzoate, methyl cinnamate, methyl salicylate) have also been proved to evoke responses in euglossine bees [39]. *Bombus terrestris* has been reported to respond to eugenol [39–41], and ethyl benzoate and melonal also elicited strong EAG responses which were not reported yet in bumblebees as we know. Floral scents highlighted the multi-functional nature of floral scent, ranging from the attraction of mutualists to the deterrence of antagonists, as well as direct defence through toxicity [42–44]. However, for bee pollinators, flower scents are always accompanied by other flower characteristics, such as food reward in natural conditions [45,46].

Our data showed that *B. terrestris* evaded these four volatile compounds of strawberry flowers, although these compounds caused a strong potential antennal response. This was somewhat surprising, as bumblebees are positive when pollinating strawberries. Other studies about strawberries also reported that the preference of *B. terrestris* for strawberry variety 'Sonata' was explained by significantly lower proportions of the GLVs (*E*)-2-hexenal, (*Z*)-3-hexenol and (*Z*)-3-hexenyl acetate being present in the scents [18]. (*Z*)-3-hexenyl acetate is known to be emitted after leaf injury and herbivory and as a signal in interplant communication [47]. It is also one of the most frequently emitted flower volatile compounds [23,48]. GLVs evoke a repellent response in agreement with the known role of this volatile in the plant's defence mechanisms. Because of the higher share of such GLVs in the floral scent of some strawberry varieties, it is plausible that they also evoke a repellent response to pollinating insects, thereby explaining the observed aversion of the tested bumblebees to this variety. Honey bees showed no obvious avoidance but also had no obvious preference. It is similar to the study of bees on tomatoes. Neither naïve

honey bees nor naïve bumblebees had a preference for tomato flower scent in a Y-tube test; however, foraging experience helped bumblebees develop a strong preference for this scent in tomato greenhouses [49]. Different concentrations of distinct volatile compounds have been reported to influence the visitation frequency of honey bees to oilseed rape and sunflowers varieties [50,51]. This might support the idea that the relative quantity of certain compounds, creating a unique blend of volatiles, might be a driver for the distinctiveness among floral scents [32,52]. However, changes in the concentration of volatile compounds are subtle, and how it affects bee pollination is still poorly understood.

## 5. Conclusions

In conclusion, honey bees and bumblebees have excellent effects in improving strawberry fruit quality. The volatile compound composition and content of strawberry 'Red face' flowers were significantly different from other varieties. These differences may be one of the factors that contribute to the foraging preference of pollinators during pollination. The EAG tests suggested that honey bees and bumblebees recognized and responded differently to compounds. Bees do not necessarily react to individual compounds in the same way as they do in practice; multiple compound combinations and subtle dosing combinations can cause different behavior responses. Our results confirmed that the response of honey bees and bumblebees to the floral scents of strawberry flowers were different. However, in the present study, not all identified volatile compounds were tested in honey bees and bumblebees. Further exploration of more compounds and field behavior observation are needed to better explain the complex interaction of strawberry pollination and provide guidance for selecting more efficient pollinator bee species or implementing more rational pollination strategies.

**Author Contributions:** Conceptualization, J.L., W.M. and Y.J.; methodology, J.L. and W.M.; software, J.L. and M.C.; validation, J.L., M.C, L.Z. and B.Z.; formal analysis, J.L. and M.C.; resources, J.L., H.Z., W.M. and Y.J.; data curation, J.L., M.C., W.M. and Y.J.; writing—original draft preparation, J.L. and M.C.; writing—review and editing, J.L., H.Z., W.M. and Y.J.; visualization, J.L.; supervision, W.M. and Y.J.; project administration, W.M.; funding acquisition, W.M. All authors have read and agreed to the published version of the manuscript.

**Funding:** This research was funded by the China Agriculture Research System (Honeybee) (CARS-44-KXJ22) and Shanxi Basic Research Program (202103021224150).

**Institutional Review Board Statement:** Not applicable.

**Informed Consent Statement:** Not applicable.

**Data Availability Statement:** The raw data supporting the conclusions of this article will be made available by the authors, without undue reservation.

**Acknowledgments:** We would like to thank Jiaxing Huang for providing the photo of bumblebee in the strawberry greenhouse. We also thanks for Wenting Su from our laboratory and Ruirui Zheng form the College of Agriculture for their help in data collection and analysis in the EAG experiment.

**Conflicts of Interest:** The authors declare that the research was conducted in the absence of any commercial or financial relationships that could be construed as a potential conflict of interest.

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
