# Peer review of "Composition of Strawberry Flower Volatiles and Their Effects on Behavior of Strawberry Pollinators, Bombus terrestris and Apis mellifera"

_agronomy, doi:10.3390/agronomy13020339_

Round 1

Reviewer 1 Report (Previous Reviewer 1)

1. The English need improvement since there are some grammatical and syntax errors in the manuscript. For example,

·         in line number 26, the words “have reported” may be as “have been reported”;

·         in line number 55, “buzzing” as “the buzzing”;

·         in line number 55 and 58, “honey” as “the honey”;

·         in line number,  “great” as “a great”;

·         in line number 76, “about how” as “how”;

·         in line number 89, “were cultivated” as “was cultivated”;

·         in line number 93, “darkroom” as “the darkroom”;

·         in line number 104, “sample” as “the sample”;

·         in line number 104, “the temperature” as “a temperature”;

·         in line number 107, “inlet” as “the inlet”;

·         in line number 109, “flower” as “the flower”;

·         in line number 112, “initial” as “the initial”;

·         in line number 154, “maintained” as “was maintained”;

·         in line number 155, “internal” as “the internal”;

·         in line number 157, “finally connect them” as “finally, connect”;

·         in line number 168, “choices” as “the choices”;

·         in line number 177, “different” as “a different”;

·         in line number 177, “chi-squared” as “the chi-squared”;

·         in line number 237, “except” as “except for”;

·         in line number 264, “greenhouse” as “the greenhouse”;

·         in line number 276,  “variety” as “the variety”;

·         in line number 298, “also be proved” as “have also been proved”;

·         in line number 331, “foraging” as “the foraging”.

The grammar mistakes which are not mentioned here are also to be checked and corrected properly.

2. There are some typing mistakes as well, and authors are advised to carefully proof-read the text. For example,

·         in line number 57, the word “driver” may be as “drivers”;

·         in line number 57, “decision” as “decisions”;

·         in line number 74, “we have” as “We have”;

·         in line number 76,  “strawberry” as “strawberries”;

·         in line number 77, “relies” as “rely”;

·         in line number 85, “driven” as “driving”;

·         in line number 105, “UAS” as “USA”;

·         in line number 109, “sample” as “samples”;

·         in line number 111, “C;helium” as “C; helium”;

·         in line number 140 an 159, “air flow” as “airflow”;

·         in line number 265, “influence tha” as “influences that”;

·         in line number 265, “strawberries” as “strawberry”;

·         in line number 287, “difference” as “differences”;

·         in line number 287, “clearly” as “clear”;

·         in line number 313, “these volatiles” as “this volatiles”.

The typos not mentioned here are also to be checked and corrected properly.

3. The authors should be checked carefully the abbreviations used in the manuscript. Check the abbreviations throughout the manuscript and introduce the abbreviation when the full word appears the first time in the abstract and the remaining for the text and then use only the abbreviation (For example, in line number 17, the authors used  both for gas chromatography-mass spectrometry (GC-MS) but in line number 80, only abbreviation is given GC-MS (in the rest of the manuscript), similarly, in line number 17 the authors used only abbreviations, but in the next line (18) both forms used). This has to be checked properly for other abbreviation used in the manuscript.

4. The full form of the species should be given when the first time appears in both the abstract and in the remaining part of the manuscript and it should be followed by only the first letter of the genus (For example, in line number 46 the full form the genus name is not give for Apis mellifera). Similary, in line number 228 the full form is given for “Bombus terrestris” and in the rest of the manuscript “B. terrestris”, it should be carefully checked and corrected.

5. When referring to SPSS versions beginning from 19, authors should cite ‘IBM SPSS Statistics for Windows, version 26 (IBM Corp., Armonk, N.Y., USA)'. Similarly, it has to be checked for the another software used (if applicable)

6. While writing the significance, authors should mentioned a,b,c, as the superscript of values given.

Author Response

Dear Reviewer:
We are appreciating for giving us an opportunity to revise our manuscript. We revised the manuscript point by point with the reviewers’ comments. we have also chcked and modified grammar mistakes and some typing mistakes to improve the quality of this manuscript. All revisions made to the manuscript were marked up using the “Track Changes” function, such that changes can be easily viewed by the editors and reviewers.

Reviewer 2 Report (Previous Reviewer 2)

Based on the revised version, this manuscript is fine and can be accepted for publication.

Author Response

Dear Reviewer:
We are appreciating for giving us an opportunity to revise our manuscript. We revised the manuscript point by point with the reviewers’ comments. we have also chcked and modified grammar mistakes and some typing mistakes to improve the quality of this manuscript. All revisions made to the manuscript were marked up using the “Track Changes” function, such that changes can be easily viewed by the editors and reviewers.

Reviewer 3 Report (New Reviewer)

Dear author

This study is appropriate research on the Argonomy journal. 

Author Response

Dear Reviewer:
We are appreciating for giving us an opportunity to revise our manuscript. We revised the manuscript point by point with the reviewers’ comments. we have also chcked and modified grammar mistakes and some typing mistakes to improve the quality of this manuscript. All revisions made to the manuscript were marked up using the “Track Changes” function, such that changes can be easily viewed by the editors and reviewers.

This manuscript is a resubmission of an earlier submission. The following is a list of the peer review reports and author responses from that submission.

Round 1

Reviewer 1 Report

1. The English need improvement since there are some grammatical and syntax errors in the manuscript. For example,

·         in line number 13, the words “the strawberry” may be as “strawberry”;

·         in line number 21, “reported” as “been reported”;

·         in line number 37, “of bumblebee” as “on bumblebee”;

·         in line number 38, “more” as “is more”;

·         all over the manuscript, “greenhouse” as “the greenhouse”;

·         in line number 56, “buzzing” as “the buzzing”;

·         in line number 56, “was different” as “being different”;

·         in line number 56, “honey” as “the honey”;

·         in line number 74 and 319, “foraging” as “the foraging”;

·         in line number 87, “main” as “the main”;

·         in line number 87, “strawberry” as “the strawberry”;

·         in line number 90, “a 100” as “100”;

·         in line number 213, “company” as “the company”;

·         in line number 116, “active” as “the active”;

·         in line number 116, “, a scalpel” as “, a and scalpel”;

·         in line number 117, “inserted a” as “inserted”;

·         in line number 120, “inserted a” as “inserted into a”;

·         in line number 149, “before air” as “before the air”;

·         in line number 153, “the arms” as “arms”;

·         in line number 159, “of choices” as “in choices”;

·         in line number 161, “of total” as “of the total”;

·         in line number 177, “in total” as “of total”;

·         in line number 207, “to same” as “to the same”;

·         in line number 208, “significantly” as “were significantly”;

·         in line number 215, “of same” as “of the same”;

·         all over the manuscript, “at concentration” as “at a concentration”;

·         all over the manuscript, “had significant” as “had a significant”;

·         in line number 226, “except” as “were except”;

·         in line number 258, “have” as “has”;

·         in line number 277, “, almost” as “, and almost”;

·         in line number 286, “proved” as “proved to”;

·         in line number 287, “respond” as “to respond”;

·         in line number 291, “always” as “are always”;

·         in line number 294, “strong” as “a strong”;

·         in line number 324, “flowers was” as “flowers were”. 

The grammar mistakes which are not mentioned here are also to be checked and corrected properly.

2. There are some typing mistakes as well, and authors are advised to carefully proof-read the text. For example,

·         in line number 33, the words “one third” may be as “one-third”;

·         in line number 39, “vegetable” as “vegetables”;

·         in line number 67, “exhibit” as “exhibits”;

·         in line number 74,  “decision” as “decisions”;

·         in line number 77, “understanding” as “understand”;

·         in line number 212, “catch” as “caught”;

·         in line number 132, “interval” as “intervals”;

·         in line number 177, “accounted” as “accounting”;

·         in line number 208, “3).The” as “3). The”;

·         in line number 215, “indicating” as “indicate”;

·         in line number 235, “none significantly” as “no significantly”;

·         in line number 209, “known” as “know”. 

The typos not mentioned here are also to be checked and corrected properly.

3. Check the abbreviations throughout the manuscript and introduce the abbreviation when the full word appears the first time in the abstract and the remaining for the text and then use only the abbreviation (For example, GC-MS, etc.,). Make a word abbreviated in the article that is repeated at least three times in the text, not all words d to be abbreviated.

4. The full form of the species should be given when the first time appears in both the abstract and in the remaining part of the manuscript and it should be followed by only the first letter of the genus (e.g., Bombus terrestris when the first time appears and followed by B. terrestris).

5. There is no software and version is given in the statistical analysis and it should be given properly in the materials and methods under the heading “Statistical analysis”.

Author Response

Dear Reviewer:

We are appreciating for giving us an opportunity to revise our manuscript. We revised the manuscript point by point with your comments. We have also modified more sentences to improve the quality of our MS. All revisions made to the manuscript were marked up using the “Track Changes” function in our MS Word, such that changes can be easily viewed by the editors and reviewers. The response was as following:

Reviewer 2 Report

Please find the comment on the manuscript in the PDF file attachment or below.

Manuscript ID: agronomy-2099042-peer-review-v1

Major revision is required.

Comments are below…

1. What is the reason behind the study? In general, the strawberry flower does not popular for food applications or other related fields. Please justify the benefit of finding out about this study from their volatile compound or their application. 

2. Typographical errors exist throughout the manuscript. Rectify carefully.

3. The authors should be more concerned the plagiarisms of the writing. Try to modify the sentence following the PDF evident attachment file form the plagiarisms checking software.

4. The collection or condition of strawberry flower harvesting or storage (in term of preparation before further analysis) needs to provide in the material and method section

5. For introduction part, Acceptable, but it would be good to highlight the limitations of previous studies and how your study can help to bridge the research gap.

6. In lines 268-279, the authors try to show the different compounds in the current study when compared to the previous study, however, the author found a differing point of their compounds but there is no reason for explaining the different points. Please find out or discuss more (1 or 2 sentences) for giving the possibility of the present results.

Author Response

(The authors gave the same response as above.)
